# Anti-Epileptic Effects of FABP3 Ligand MF1 through the Benzodiazepine Recognition Site of the GABA_A_ Receptor

**DOI:** 10.3390/ijms21155525

**Published:** 2020-08-01

**Authors:** Yasushi Yabuki, Jiaqi Liu, Ichiro Kawahata, Hisanao Izumi, Yasuharu Shinoda, Kohei Koga, Shinya Ueno, Norifumi Shioda, Kohji Fukunaga

**Affiliations:** 1Department of Pharmacology, Graduate School of Pharmaceutical Sciences, Tohoku University, Sendai 980-8578, Japan; yabukiy@kumamoto-u.ac.jp (Y.Y.); liu.jiaqi.r4@dc.tohoku.ac.jp (J.L.); kawahata@tohoku.ac.jp (I.K.); kfukunaga@tohoku.ac.jp (H.I.); yshinoda@tohoku.ac.jp (Y.S.); 2Department of Genomic Neurology, Institute of Molecular Embryology and Genetics, Kumamoto University, Kumamoto 860-0811, Japan; shioda@kumamoto-u.ac.jp; 3Department of Neurophysiology, Hyogo College of Medicine, Nishinomiya 663-8501, Japan; kkoga@hotmail.co.jp; 4Department of Neurophysiology, Graduate School of Medicine, Hirosaki University, Hirosaki 036-8216, Japan; shinyau@hirosaki-u.ac.jp

**Keywords:** GABA_A_ receptor, anti-epileptic effect, benzodiazepine recognition site

## Abstract

Recently, we developed the fatty acid-binding protein 3 (FABP3) ligand MF1 (4-(2-(1-(2-chlorophenyl)-5-phenyl-1H-pyrazol-3-yl)phenoxy) butanoic acid) as a therapeutic candidate for α-synucleinopathies. MF1 shows affinity towards γ-aminobutyric acid type-A (GABA_A_) receptor, but its effect on the receptor remains unclear. Here, we investigate the pharmacological properties of MF1 on the GABA_A_ receptor overexpressed in Neuro2A cells. While MF1 (1–100 μm) alone failed to evoke GABA currents, MF1 (1 μm) promoted GABA currents during GABA exposure (1 and 10 μm). MF1-promoted GABA currents were blocked by flumazenil (10 μm) treatment, suggesting that MF1 enhances receptor function via the benzodiazepine recognition site. Acute and chronic administration of MF1 (0.1, 0.3 and 1.0 mg/kg, p.o.) significantly attenuated status epilepticus (SE) and the mortality rate in pilocarpine (PILO: 300 mg/kg, i.p.)-treated mice, similar to diazepam (DZP: 5.0 mg/kg, i.p.). The anti-epileptic effects of DZP (5.0 mg/kg, i.p.) and MF1 (0.3 mg/kg, p.o.) were completely abolished by flumazenil (25 mg/kg, i.p.) treatment. Pentylenetetrazol (PTZ: 90 mg/kg, i.p.)-induced seizures in mice were suppressed by DZP (5.0 mg/kg, i.p.), but not MF1. Collectively, this suggests that MF1 is a mild enhancer of the GABA_A_ receptor and exercises anti-epileptic effects through the receptor’s benzodiazepine recognition site in PILO-induced SE models.

## 1. Introduction

γ-aminobutyrate (GABA) is a critical inhibitory neurotransmitter in both the mammalian peripheral and central nervous systems and maintains brain homeostasis by mediating neuronal excitability [1,2]. A collapse of GABAergic transmission leads to neuronal hyperexcitability and the subsequent development of epileptic seizures [3,4]. An aberrant GABAergic system is also associated with neurological disorders such as Alzheimer’s disease (AD), Parkinson’s disease (PD) and autism spectrum disorders (ASD) [5,6,7], often co-occurring with epileptic seizures [8,9]. Many anticonvulsants promote GABAergic neuronal transmission by enhancing the activity of the GABA type-A (GABA_A_) receptor and increasing GABA content in the synaptic cleft [4,10]. However, 20–30% of patients with epilepsy showed resistance to existing anticonvulsants [11]. Therefore, the development of powerful novel and safe anticonvulsants are necessary to improve the quality of life for patients with epilepsy.

The GABA_A_ receptor complex consists of five subunits of different families (α1–6, β1–4, γ1–4, δ, ϵ and π) [12]. In the mammalian brain, the αβγ-subunit-containing complex is the major component of the GABA_A_ receptor; a minor complex with αβδ subunits is also observed in the hippocampal dentate gyrus, thalamus and cerebellum [13]. The GABA_A_ receptor function is dependent on the type of subunit complex. For instance, the αβγ complex is sensitive to benzodiazepine, but this is not so for the αβδ complex [12,13]. The α1βγ-type GABA_A_ receptors are localized in the post synaptic region, while α4βδ-type GABA_A_ receptors are observed in extra-synaptic region [14,15].

Fatty acid-binding proteins (FABPs) are essential for the intracellular trafficking of long-chain polyunsaturated fatty acids and mediating their transport to the plasma membrane, mitochondria and nucleus [16]. Of the twelve types of FABP, FABP3, FABP5 and FABP7 are predominantly expressed in human and rodent brains [17,18]. FABP3 has been proposed as a potential biomarker of neurodegeneration, as increased levels of FABP3 have been observed in the cerebrospinal fluid (CSF) of patients with AD, PD, and vascular dementia [19,20,21]. We previously demonstrated that FAPB3 has a critical role in the development and progression of α-synucleinopathies such as PD and dementia with Lewy bodies (DLB). FABP3 null mice show resistance to 1-methyl-4-phenyl-1,2,3,6-tetrahydropyridine (MPTP)-induced dopaminergic neuronal toxicity and PD-like motor deficits [22]. FABP3 deficiency also prevents the spreading of α-synucleinopathy-like pathologies following the injection of α-synuclein preformed fibril (PFF) in the mouse brain [23] and uptake into dopaminergic neurons [24]. Moreover, we have succeeded in producing a novel FABP3 ligand, MF1 (4-(2-(1-(2-chlorophenyl)-5-phenyl-1H-pyrazol-3-yl)phenoxy) butanoic acid), which has a high affinity for FABP3 [25,26]. MF1 significantly attenuates loss of dopaminergic neurons, behavioral impairments and α-synucleinopathy-like pathologies observed in MPTP-treated and/or α-synuclein PFF injected mice [23,26]. These results suggest that FABP3 is a potential molecular target producing novel therapeutics for α-synucleinopathies. Our off-target analysis showed that MF1 has an affinity for the GABA_A_ receptor, but its effect on the GABAergic system remains unclear.

Here, we test whether MF1 affects GABA_A_ receptor function using the whole cell patch-clamp technique on Neuro2A cells that overexpress the GABA_A_ receptor. We also validate the anti-epileptic effect of MF1 in pilocarpine (PILO)-induced status epilepticus (SE) and pentylenetetrazol (PTZ)-induced epilepsy in mice. Our results suggest that MF1 is able to enhance the GABA_A_ receptor activity through the benzodiazepine recognition site, thereby demonstrating anticonvulsant-like effects.

## 2. Results

### 2.1. MF1 Promotes GABA_A_ Receptor Currents through the Benzodiazepine Recognition Site

First, we evaluated the effect of MF1 on the GABA_A_ receptor. We confirmed that the application of GABA (1–100 μm) induces inward currents, indicating that the overexpressed GABA_A_ receptors function normally in Neuro2A cells (Figure 1A,C). However, application of MF1 (1–100 μm) alone did not evoke GABA currents (Figure 1B,C). Next, we tested whether MF1 affects GABA-induced currents. Significant group effects were observed upon application of 1 μm GABA [F(2, 18) = 28.12, *p* < 0.0001] and 10 μm GABA [F(2, 18) = 14.43, *p* = 0.0002], with or without treatment with MF1 (1 μm) and flumazenil (10 μm) (Figure 1D,E). Co-application of MF1 (1 μm) markedly increased GABA (1 - 10 μm) currents (1 μm: 2.2 ± 0.07, *p* < 0.01 vs. application of GABA alone; 10 μm: 1.4 ± 0.07, *p* < 0.01 vs. application of GABA alone; Figure 1D,E). Since benzodiazepines facilitate GABA_A_ currents without evoking them [27], we speculated that MF1 promotes GABA_A_ currents in a manner similar to benzodiazepines. As expected, flumazenil (10 μm), which inhibits the GABA_A_ receptor by binding the benzodiazepine recognition site, significantly blocking the MF1-promoted GABA current (1 μm: 1.2 ± 0.1, *p* > 0.05 vs. application of GABA alone, *p* < 0.01 vs. application of GABA and MF1; 10 μm: 0.82 ± 0.08, *p* > 0.05 vs. application of GABA alone, *p* < 0.01 vs. application of GABA and MF1; Figure 1D,E).

### 2.2. Acute MF1 Administration Attenuates SE and Mortality in PILO-Treated Mice

Next, we assessed the anti-epileptic effect of MF1 using animal models of epilepsy. Animal experimental schedules are shown in Figure 2A. We utilized diazepam (DZP: 5.0 mg/kg, i.p.) as a positive control [28,29]. We observed significant group effects on SE onset time [F(4, 66) = 24.87, *p* < 0.0001] and Racine scale score [F(4, 98) = 7.425, *p* < 0.0001]. Likewise, DZP (5.0 mg/kg, i.p.) treatment and acute administration of MF1 (0.3 and 1.0 mg/kg, p.o.) significantly prolonged SE onset time (0.3 mg/kg: 14.6 ± 1.2 min, *p* < 0.01 vs. PILO-treated mice; 1.0 mg/kg: 13.9 ± 0.8 min, *p* < 0.01 vs. PILO-treated mice; Figure 2B) and reduced the Racine scale score (0.3 mg/kg: 3.3 ± 0.3, *p* < 0.05 vs. PILO-treated mice; 1.0 mg/kg: 3.3 ± 0.2, *p* < 0.01 vs. PILO-treated mice; Figure 2C) in PILO-treated mice.

Administration of MF1 (0.3 and 1.0 mg/kg, p.o.) also extended the survival rate over 90 min (0.3 mg/kg: 89%, *p* < 0.01 vs. PILO-treated mice; 1.0 mg/kg: 89%, *p* < 0.01 vs. PILO-treated mice; Figure 2D) and 7 days (0.3 mg/kg: 72%, *p* < 0.05 vs. PILO-treated mice; 1.0 mg/kg: 89%, *p* < 0.01 vs. PILO-treated mice; Figure 2D) after PILO (300 mg/kg, i.p.) injection and treatment with DZP (5.0 mg/kg, i.p.) (Figure 2D).

### 2.3. Acute MF1 Administration Does Not Improve Epileptic Seizures in PTZ-Treated Mice

The anti-epileptic effect of MF1 was also investigated using PTZ-treated mice. DZP (5.0 mg/kg, i.p.) significantly suppressed the onset time of generalized tonic-clonic seizures (GTCs) and mortality, as compared to vehicle-administered PTZ-treated mice. However, MF1 had no effect in PTZ-treated mice (Table 1), suggesting that the anti-epileptic effect of MF1 (0.1–1.0 mg/kg, p.o.) may be milder than that of DZP (5.0 mg/kg, i.p.).

### 2.4. Chronic MF1 Administration Suppresses SE and Mortality in PILO-Treated Mice

We also investigated whether the chronic administration of MF1 rescues epileptic seizures in PILO-treated mice. Animals were treated with PILO (300 mg/kg, i.p.) 24 h after final MF1 and DZP administration (Figure 2A). A significant group effect was observed on SE onset time [F(4, 95) = 5.922, *p* = 0.0003], but not on the Racine scale score [F(4, 136) = 2.152, *p* = 0.0778]. Whereas chronic DZP (5.0 mg/kg, i.p.) or MF1 administration significantly prolonged SE onset time (0.1 mg/kg: 12.7 ± 0.68 min, *p* < 0.01 vs. PILO-treated mice; 0.3 mg/kg: 12.0 ± 0.79 min, *p* < 0.05 vs. PILO-treated mice; 1.0 mg/kg: 12.7 ± 0.91 min, *p* < 0.01 vs. PILO-treated mice; Figure 3A), both drugs failed to lower the Racine scale score in PILO-treated mice (Figure 3B). Chronic administration of DZP (5.0 mg/kg, i.p.) or MF1 (0.1 mg/kg, p.o.) significantly improved the survival rate over 90 min (0.1 mg/kg: 19%, *p* < 0.05 vs. PILO-treated mice; Figure 3C) and 7 days (0.1 mg/kg: 22%, *p* < 0.01 vs. PILO-treated mice; Figure 3C) after PILO (300 mg/kg, i.p.) injection. MF1 (0.3 and 1.0 mg/kg) slightly improved the survival rate, but the effect was not significant (Figure 3C).

### 2.5. Flumazenil Blocks Anti-Epileptic Effect of MF1 in PILO-Treated Mice

Finally, we confirmed whether the acute administration of MF1 (0.3 mg/kg, p.o.) shows an anti-epileptic effect by enhancing the GABA_A_ receptor through the benzodiazepine recognition site. We observed significant group effects on the SE onset time [F(4, 70) = 29.54, *p* < 0.0001] and Racine scale score [F(4, 106) = 11.89, *p* < 0.0001]. Consistent with results of the whole cell patch-clamp recording, flumazenil (25 mg/kg, i.p.) significantly prevented the acute effect of MF1 (0.3 mg/kg, p.o.) on the SE onset time (9.5 ± 2.0 min, *p* > 0.05 vs. PILO-treated mice, *p* < 0.01 vs. MF1-administered PILO-treated mice; Figure 4A) and Racine scale score (3.9 ± 0.40, *p* > 0.05 vs. PILO-treated mice, *p* = 0.095 vs. MF1-administered PILO-treated mice; Figure 4B) in PILO-treated mice. The improvement of the survival rate by MF1 (0.3 mg/kg, p.o.) was also inhibited by flumazenil (25 mg/kg, i.p.) treatment (55.6%, *p* > 0.05 vs. PILO-treated mice, *p* < 0.01 vs. MF1-administered PILO-treated mice; Figure 4C). Flumazenil (25 mg/kg, i.p.) also inhibited the effect of DZP (5.0 mg/kg, i.p.) in PILO-treated mice (Figure 4A–C). Therefore, MF1 may exercise its anti-epileptic effects by promoting GABA_A_ receptor function in a manner similar to DZP.

## 3. Discussion

In the present study, we demonstrated that the FABP3 ligand MF1 promotes GABA currents in GABA_A_ receptor-expressing cells. MF1 also exhibits anti-epileptic effects on PILO-induced SE in mice. These effects of MF1 were blocked by treatment with flumazenil, suggesting that MF1 enhances GABA_A_ receptor function directly via the benzodiazepine recognition site.

Here, we show that MF1 enhances GABA currents arising from the GABA_A_ receptor complex with α1β2γ2 subunits, which is the major complex found in the brain (approximately 60% of all GABA_A_ receptors) [13]. However, the effects of MF1 on other types of GABA_A_ receptors is unclear. Since MF1-promoted GABA currents were completely blocked by flumazenil, we propose that MF1 enhances the function of at least the αβγ-type benzodiazepine sensitive GABA_A_ receptor.

Benzodiazepine receptor agonists act on not only the GABA_A_ receptor, but also delayed rectifier K^+^ channels. Midazolam behaves like an open-channel inhibitor with a rapid onset of blocking and without frequency dependence on the block for the delayed rectifier K^+^ channels, thereby inhibiting the amplitude of action potentials in NSC-34 neuronal cells [30,31]. This effect is not blocked by flumazenil [31], suggesting that the benzodiazepine receptor agonist has the possibility of suppressing neuronal excitabilities, in vivo and independently of GABAergic action. As the MF1 here shows benzodiazepine receptor agonist-like effects, it may also affect the electrophysiological properties of delayed rectifier K^+^ channels. We will try to investigate the effect of MF1 on the delayed rectifier K^+^ currents in a future study.

Acute DZP (5.0 mg/kg, i.p.) treatment significantly attenuated the epileptic seizures observed in both PILO- and PTZ-treated mice. However, the acute administration of MF1 (0.3 and 1.0 mg/kg, p.o.) failed to inhibit PTZ-induced GTCs and mortality. Assuming that the anti-epileptic effects of DZP and MF1 arise from similar mechanisms of action, given that they both bind to the benzodiazepine recognition site, this result suggests that DZP may be a more powerful enhancer of the GABA_A_ receptor function than MF1 in mouse brain. PILO leads to aberrant excitability, followed by the generation of seizures by stimulating the muscarinic acetylcholine receptor [32,33,34], while PTZ induces epileptic seizures by antagonizing GABA_A_ receptor via the picrotoxinin-sensitive site [35,36]. Thus, acute MF1 (1.0 mg/kg, p.o.) administration appears to be insufficient to enhance the activity of the GABAergic system in the background of the impaired GABA_A_ receptor function in PTZ-treated mice. However, a recent report indicates that fatty acid amides, with an affinity for FABPs, show anti-epileptic action in PTZ-treated mice, an effect blocked by flumazenil [37]. Further studies are necessary to investigate the effect of MF1 on other types of GABA_A_ receptors.

The chronic administration of MF1 also attenuated PILO-induced SE and mortality. Since MF1 has a long half-life of not less than 20 h, the concentration of MF1 in the brain by chronic administration is sufficient to enhance GABA_A_ receptor activity in the background of PILO treatment. We have previously reported that deletion of FABP3 upregulates GABAergic transmission by increasing the expression levels of the GABA, synthesizing enzyme glutamic acid decarboxylase 67 (GAD67), thereby suppressing excitability in the mouse anterior cingulate cortex [38,39]. Hence, chronic administration of MF1 may also promote the GABAergic system activity in the cortex by increasing GAD67 expression. We plan to assess the chronic effects of MF1 on GAD67 levels in the future.

While DZP is the drug of choice for the treatment of early SE, for acute repetitive seizures and febrile seizure prophylaxis [40,41], it has many side effects such as somnolence, depression, nausea, motor coordination disorder, and dizziness [41]. Moreover, repeated DZP treatment leads to the development of benzodiazepine tolerance and withdrawal syndrome, making it unsuitable for long-term epilepsy therapy [40,41]. Chronic administration (at least once a day for seven consecutive weeks) of MF1 (1.0 mg/kg, p.o.) did not affect motor and cognitive function in mice [23,26]. While the tolerance and withdrawal of MF1 should be analyzed, we hope that MF1 may be a safer alternative for long-term therapy.

We originally developed MF1 as therapeutic candidate for α-synucleinopathies [23,26]. MF1 attenuates aggregation and spreading of α-synuclein by preventing interactions between FABP3 and α-synuclein [23,26], suggesting that MF1 has the potential for early treatment of α-synucleinopathies. Patients with DLB cause seizures (14.7%) and myoclonus (58.1%) after disease onset, indicating that the seizure incidence rates are approximately 10-fold, relative to healthy controls [42,43]. Therefore, MF1 may be effective in alleviating not only core symptoms, but peripheral symptoms as well.

In conclusion, we have highlighted the pharmacological properties of MF1 on the GABAergic inhibitory system. Like the benzodiazepine agent DZP, MF1 may enhance function of the GABA_A_ receptor, and in turn, suppress epileptic seizures in PILO-treated mice. Therefore, we suggest MF1 as an attractive therapeutic candidate for neurodegenerative disorders with epilepsy, including α-synucleinopathies.

## 4. Materials and Methods

### 4.1. Animals

Male ICR mice were purchased from Clea Japan, Inc. (Tokyo, Japan). Adult male mice (8–10 weeks old, weight 30–45 g) were used for all the experiments. Animals were housed under conditions of constant temperature 23 ± 2 °C and humidity 55 ± 5%, in a 12 h light–dark cycle (light: 9 am–9 pm). The mice had free access to food and water. All experimental procedures using animals were approved (2019 PhA-024, 1st April 2019) by the Committee on Animal Experiments at Tohoku University. We made an effort to reduce animal suffering and to use the minimum number of mice.

### 4.2. Chemicals

For administration to the animals, scopolamine hydrobromide (cholinergic inhibitor), DZP, PILO (muscarinic cholinergic agonist), and PTZ (GABA_A_ receptor antagonist) (Sigma-Aldrich, St Louis, MO, USA) were dissolved in saline (0.9% NaCl). Flumazenil (Wako Pure Chemicals, Osaka, Japan) was prepared with 0.05% Tween-80 in saline. MF1 was synthesized as described previously [44] and suspended in 0.5% carboxymethylcellulose sodium salt (CMC). Doses of DZP (5.0 mg/kg, i.p.) and flumazenil (25 mg/kg, i.p.) were chosen, since these doses are enough to work in vivo without the influence of normal animal conditions [28].

For patch-clamp recording, GABA (Wako Pure Chemicals) was dissolved in distilled water. MF1 and flumazenil were suspended in dimethyl sulfoxide (DMSO) to achieve final concentrations of 0.1–0.01% for the assay.

### 4.3. Cell Transfection

Neuro2A mouse neuroblastoma cells were incubated in Dulbecco’s minimal essential medium (DMEM) with 10% heat-inactivated fetal bovine serum (FBS) and penicillin/streptomycin (100 units/100 μg/mL) in a 5% CO_2_ incubator at 37 °C. Cells were transfected with pcDNA plasmids (1.0 μg/mL), encoding the rat GABA_A_ receptor subunits α1, β2 and γ2, with 0.1% green fluorescent protein (GFP) plasmid, by lipofection, as previously described [45]. GABA_A_ receptor-expressing cells were used for electrophysiological recording 24–48 h after transfection.

### 4.4. Whole Cell Patch-Clamp Recording

GABA_A_ receptor currents were recorded as previously described [45,46]. The external solution contained 143 mM NaCl, 5 mM KCl, 2 mM CaCl_2_, 1 mM MgCl_2_, 10 mM glucose and 10 mM HEPES (Tyrode’s solution; pH adjusted to 7.4 with NaOH). Glass pipettes were filled with internal solution containing 140 mM CsCl, 2 mM Mg-ATP, 10 mM EGTA, 10 mM HEPES (pH adjusted to 7.4 with CsOH). The resistance of electrodes filled with internal solution was 3.0–4.5 MΩ. Rapid drug application was achieved using the ALA-VM4 system (Sutter Instrument Company, Novato, CA, USA). GABA_A_ receptor currents were recorded at room temperature using an EPC10 single patch clamp amplifier and acquisition system (HEKA, Lambrecht, Germany), filtered at 3 kHz, and sampled at 10 kHz. The membrane potential was clamped at −60 mV. Measured GABA currents were normalized to membrane capacitance in each cells.

### 4.5. Evaluation of Epileptic Behaviors

To assay the anti-epileptic effect of MF1, we utilized PILO- and PTZ-induced epileptic mice. To assess the acute drug effects, some animals were treated with MF1 (0.1, 0.3 and 1.0 mg/kg, p.o.) and DZP (5.0 mg/kg, i.p.) 30 min before injection with PILO (300 mg/kg, i.p.) or PTZ (90 mg/kg, i.p.). The competitive inhibitor of GABA_A_ receptor, flumazenil (25 mg/kg, i.p.) was administered 5 min prior to treatment with MF1 (0.3 mg/kg, p.o.) and DZP (5.0 mg/kg, i.p.). To assess the chronic drug effects, the same doses of MF1 and DZP were administered once a day for seven consecutive days before PILO injection. Animal experimental schedules are shown in Figure 2A.

#### 4.5.1. PILO-Induced SE Model

To suppress the peripheral cholinergic side effects, mice were pre-treated with scopolamine (1.0 mg/kg, i.p.) 30 min prior to PILO (300 mg/kg, i.p.) injection. Each mouse was placed in a plastic cage and its behavior was recorded for 90 min after PILO injection. The seizure stages of the mice were evaluated, based on the following Racine scale score (1972) [47]: 0- No behavioral seizures; 1-Mouth and facial movements; 2-Head myoclonus; 3-Forelimb myoclonus; 4-Forelimb myoclonus followed by rearing; 5-Falling or generalized tonic–clonic convulsions. If mice died during these 90 min, the Racine scale score was assigned as stage 5. Stages 3–5 are defined as convulsive seizures [48,49]. Mice that exhibited intermittently persistent stage 3–5 seizures at least three times within 90 min of PILO injection were considered to undergo SE. SE onset time was measured as the time of the first observation of convulsive seizures. To terminate SE, all the animals were treated with DZP (5 mg/kg, i. p.), which was repeated, if needed, to suppress convulsions. To minimize suffering, moistened rodent chow and an injection of 2 mL of 5% glucose were given to all the mice for 5 days after SE.

#### 4.5.2. PTZ-Induced Seizure

Mouse behavior was monitored and recorded in individual cages for 30 min after injection of PTZ (90 mg/kg, i.p.). We scaled seizure behavior as follows (modified from Racine, 1972) [47]: 1-Hypoactivity; 2-Tail extension or limb jerk; 3-Whole-body clonus; 4-Rolling, running, jumping, and tonic-clonic. GTCs were assigned as stage 4.

### 4.6. Statistical Analysis

Data are shown as mean ± standard error of the mean (SEM). Significant differences were determined using Student’s t-test for two-group comparisons or a one-way analysis of variance (ANOVA) for multi-group comparisons, followed by Bonferroni’s multiple comparison test. Statistically significant differences of Kaplan Meier survival curves were tested by log-rank test using GraphPad Prism 7.04 (GraphPad Software, Inc., La Jolla, CA, USA). *p* < 0.05 represented a statistically significant difference.

## Figures and Tables

**Figure 1 ijms-21-05525-f001:**
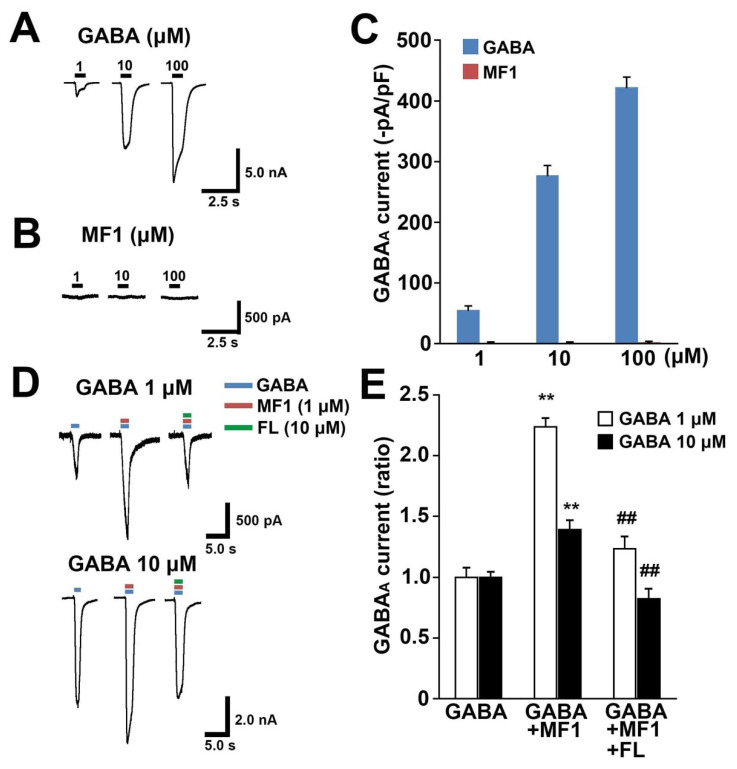
Effects of MF1 on GABA currents in GABA_A_ receptor-expressing cell. (**A**,**B**) Representative traces treated with (**A**) GABA (1–100 μm) and (**B**) MF1 (1–100 μm). (**C**) Peak current observed by treatment with GABA (1–100 μm) and MF1 (1–100 μm) was normalized to membrane capacitance (GABA: *n* = 10 per group; MF1: *n* = 8 per group). Error bars represent SEM. (**D**) Representative traces from cells treated with GABA (1 and 10 μm) in combination with MF1 (1 μm) and/or flumazenil (10 μm). (**E**) MF1 (1 μm) enhanced GABA (1 and 10 μm) currents, which were blocked by flumazenil (10 μm) (*n* = 7 per group). Error bars represent SEM. ** *p* < 0.01 vs. each GABA alone treatment. ^##^
*p* < 0.01 vs. each combination treatment of GABA and MF1. FL: flumazenil.

**Figure 2 ijms-21-05525-f002:**
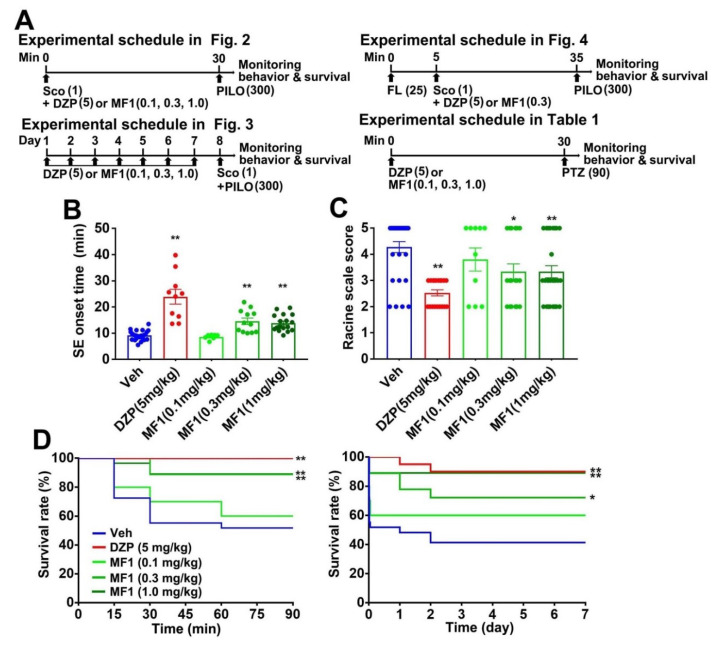
Acute effect of MF1 and diazepam (DZP) on epileptic seizure in PILO-treated mice. (**A**) Animal experimental schedules of the present study. (**B**,**C**) Effects of MF1 (0.1, 0.3 and 1.0 mg/kg, p.o.) and DZP (5.0 mg/kg, i.p.) on (**B**) SE onset time and (**C**) Racine scale score in PILO-treated mice. Error bars represent SEM. (**D**) Survival rates over 90 min (left) and 7 days (right) after PILO treatment are shown. (Vehicle: *n* = 29; DZP: *n* = 18; MF1 [0.1 mg/kg]: *n* = 10; MF1 [0.3 mg/kg]: *n* = 18; MF1 [1 mg/kg]: *n* = 27). * *p* < 0.05, ** *p* < 0.01 vs. vehicle-administered PILO-treated mice.

**Figure 3 ijms-21-05525-f003:**
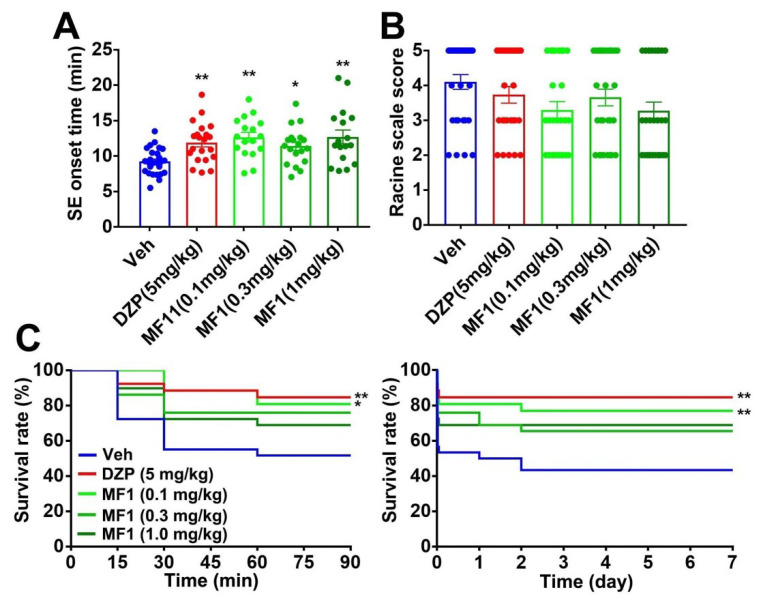
Effects of chronic MF1 and DZP administration on epileptic seizure in PILO-treated mice. (**A**) Chronic administration of MF1 (0.1, 0.3 and 1.0 mg/kg, p.o.) and DZP (5.0 mg/kg, i.p.) attenuated SE onset time in PILO-treated mice. Error bars represent SEM. (**B**) Chronic administration of MF1 (0.1, 0.3 and 1.0 mg/kg, p.o.) and DZP (5.0 mg/kg, i.p.) did not affect Racine scale score in PILO-treated mice. Error bars represent SEM. (**C**) Chronic administration of MF1 (0.1 mg/kg, p.o.) and DZP (5.0 mg/kg, i.p.) improved the reduced survival rate over 90 min (left) and 7 days (right) after PILO treatment. (Vehicle: *n* = 30; DZP: *n* = 26; MF1 [0.1 mg/kg]: *n* = 27; MF1 [0.3 mg/kg]: *n* = 29; MF1 [1 mg/kg]: *n* = 29). * *p* < 0.05, ** *p* < 0.01 vs. vehicle-administered PILO-treated mice.

**Figure 4 ijms-21-05525-f004:**
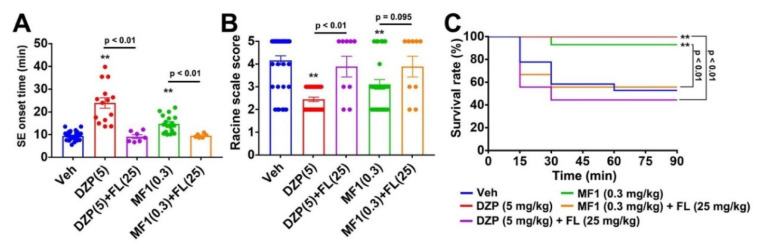
Flumazenil antagonizes the acute effects of MF1 on epilepsy in PILO-treated mice. (**A**–**C**) Treatment with flumazenil (25 mg/kg, i.p.) inhibited the effects of MF1 (0.3 mg/kg, p.o.) and DZP (5.0 mg/kg, i.p.) on (**A**) SE onset time, (**B**) Racine scale score and (**C**) survival rate in PILO-treated mice. Error bars represent SEM. (Vehicle: *n* = 36; DZP: *n* = 29; DZP + flumazenil: *n* = 9; MF1: *n* = 28; MF1 + flumazenil: *n* = 9). ** *p* < 0.01 vs. vehicle-administered PILO-treated mice. DZP (5): DZP DZP (5.0 mg/kg, i.p.) treatment; FL (25): flumazenil (25 mg/kg, i.p.) treatment; MF1 (0.3): MF1 (0.3 mg/kg, p.o.) administration.

**Table 1 ijms-21-05525-t001:** Acute effects of MF1 and DZP on epileptic seizure in PTZ-treated mice. ** *p* < 0.01 vs. vehicle-administered PTZ-treated mice. (*n* = 10 per group). GTCs: generalized tonic-clonic seizures, i.p.: intraperitoneal, p.o.: per os.

Group	GTCs Onset Time (Min)	Mortality (%)	Seizure Scale
Vehicle	1.7 ± 0.53	50 (5/10)	4 ± 0
DZP (5 mg/kg, i.p.)	3.3 ± 0.70 **	0 (0/10)	3.7 ± 0.15
MF1 (0.1 mg/kg, p.o.)	0.92 ± 0.15	40 (4/10)	3.8 ± 0.20
MF1 (0.3 mg/kg, p.o.)	1.2 ± 0.22	50 (5/10)	3.8 ± 0.20
MF1 (1.0 mg/kg, p.o.)	1.2 ± 0.28	70 (7/10)	3.8 ± 0.20

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
