# Peer review of "Anti-Epileptic Effects of FABP3 Ligand MF1 through the Benzodiazepine Recognition Site of the GABAA Receptor"

_ijms, 2020, doi:10.3390/ijms21155525_

Round 1

Reviewer 1 Report

this is a very interesting research regarding a new chemical MF1 to be potential anticonvulsive drug. HOwever, there are some questions need authors to respond and revise. 

  1. The authors claimed that they developed the FABP3 ligant MF1 as a therapeutic candidate for alpha synucleinopathies, which is highly related to some degenerative disorders, like PD. However, among different subgroups of PD, some of them indeed are related to seizures (doi: 10.1016/j.seizure.2015.02.007. Epub 2015 Feb 17.), most of PD do not have seizures clinically. Your data showed a very high responding rate (50%) for the MF1 to the SE. HOw do you explain the connection? which is not clearly, or not highlighted in your introduction or discussion.
  2. line 45 anti-anticonvulsant? 
  3. Do you think that adding a positive control of DZP in the FIgure 1 A may be more persuative? 
  4. Would it be possible that your FIgure  1 D suggesting that MF1 is a cofactor to enhance GABA activity? 
  5. Why in the in vivo study did not use GABA injection? (I guess that the answer is the half life of GABA is too short to reach brain for your experiment ?, and the intracranial injection may exacerbate the SE?)
  6. Regarding the in vivo study, why the dose of DZP is 5 mg/kg? (clinically, the dose of DZP is maximal to 1mg/kg, such a high dose is suitable?) Also, the dose of FL25 mg/kg  is too high  (annexate dose is 0.01mg/kg, maximal dose 3mg in human). Do these animal show normal response in such a high dose?  

Author Response

1) The authors claimed that they developed the FABP3 ligand MF1 as a therapeutic candidate for alpha synucleinopathies, which is highly related to some degenerative disorders, like PD. However, among different subgroups of PD, some of them indeed are related to seizures (doi: 10.1016/j.seizure.2015.02.007. Epub 2015 Feb 17.), most of PD do not have seizures clinically. Your data showed a very high responding rate (50%) for the MF1 to the SE. How do you explain the connection? which is not clearly, or not highlighted in your introduction or discussion.

Answer (Ans) Thank you for your comments. According to Beagle AJ et al (2017), the cumulative probabilities of developing seizures and myoclonus after disease onset are 14.7% and 58.1% in patients with DLB, respectively (J Alzheimers Dis. 2017, 60, 211–223.). In some cases, the patients with PD also have co-occurrences of epileptic seizure (Neurosci Res. 2001, 41, 397–399.). Therefore, we suggest that MF1 may be effective for co-occurrent epileptic seizure in patients with synucleinopathies. We described it as followed “Patients with DLB cause seizures (14.7%) and myoclonus (58.1%) after disease onset, indicating the seizure incidence rates are approximately 10-fold higher than healthy controls [42, 43].” (Line 222 to 223).

2) line 45 anti-anticonvulsant?

Ans. According to reviewer’s comment, we corrected "anti-anticonvulsants" to "anticonvulsants”. (Line 45).

3) Do you think that adding a positive control of DZP in the Figure 1 A may be more persuative?

Ans. Karim N et al reported that DZP (0.5 μM) significantly promotes GABA (10 μM)-evoked human GABAA currents (approximately 3-times upregulation) (Biochemical pharmacology. 201182, 1971–1983.). Moreover, DZP (1.0 μM) also makes GABA (3.2 μM)-evoked rat GABAA currents over 3-fold (Nature neuroscience. 20003, 1274–1281.). Therefore, we suggest that DZP (1 μM) facilitates GABA (1 and 10 μM)-evoked currents more than that of MF1 at the same dose.

4) Would it be possible that your Figure 1D suggesting that MF1 is a cofactor to enhance GABA activity?

Ans. We think it’s undeniable possibility. Whereas MF1 alone fails to produce GABA currents (Fig. 1B, C), MF1 promotes the GABA-evoked currents in GABAA receptor-expressed cells, an effect blocked by flumazenil application (Fig. 1D). Therefore, we here concluded that MF1 show anti-epileptic effects by enhancing GABAA receptor though the benzodiazepine recognition site.

5) Why in the in vivo study did not use GABA injection? (I guess that the answer is the half life of GABA is too short to reach brain for your experiment?, and the intracranial injection may exacerbate the SE?)

Ans. Since MF1 enhances GABAA receptor function through the benzodiazepine recognition site, we used DZP as a positive control for benzodiazepine receptor agonist, but not GABA. Shyamaladevi N et al reported that administration of GABA (600 mg/kg, i.p.) alone increases brain GABA levels (an increase of 33%) in rats (Brain Res Bull. 2002, 57, 231–236.). However, GABA at a dose of 600 mg/kg is so high concentration and there is no evidence whether GABA (600 mg/kg, i.p.) attenuates epilepsy. Generally, malfunction of GABAergic system leads to elevate excitability and then develops epileptic seizures. Since intracranial injection of GABA suppresses excitability in the brain (Nature. 1958, 182, 1076–1077.), intracranial injection may be possible to suppress SE in PILO-treated mice.

6) Regarding the in vivo study, why the dose of DZP is 5 mg/kg? (clinically, the dose of DZP is maximal to 1mg/kg, such a high dose is suitable?) Also, the dose of FL25 mg/kg is too high (annexate dose is 0.01mg/kg, maximal dose 3mg in human). Do these animal show normal response in such a high dose?  

Ans. We selected the dose of DZP and flumazenil according to previous reports (Neurosci Lett. 2012, 523, 115–118.). Since we used DZP as a positive control for anticonvulsants of benzodiazepine, we chose a sure effective dose in the present study. DZP (5.0 mg/kg, i.p.) or flumazenil (25 mg/kg, i.p.) did not alter the condition of mice (Neurosci Lett. 2012, 523, 115–118.). We also described the reason why we choose the dose of drugs as followed “Doses of DZP (5.0 mg/kg, i.p.) and flumazenil (25 mg/kg, i.p.) were chosen, since these doses are enough to work in vivo without influence on normal animal condition [28].” (Line 243 to 245)

Reviewer 2 Report

The present results showed that, similar to the benzodiazepine agent diazepam, MF1 (4-(2-(1-(2-58 chlorophenyl)-5-phenyl-1H-pyrazol-3-yl)phenoxy) butanoic acid) might be able to enhance function of the GABA(A) receptor, and hence, to suppress epileptic seizures in pilocarpine-treated mice. The results could be interesting to some extent.

Some of comments are shown in the following:

Major comments:

  • Previous reports have demonstrated that benzodiazepine (e.g., midazolam or diazepam) might be capable of modulating the amplitude and gating of delayed rectifier K+ currents and that the underlying inhibitory action was not reversed by further addition of flumazenil (Vonderlin et al., Drug Des Devel Ther 2014;8:2263-2271; So et al., Eur J Pharmacol 2014;724:152-160). The delayed rectifier K+ currents are present in Neuro2A cells and they are highly likely to be inhibited by benzodiazepine derivatives (e.g., MF1, [4-(2-(1-(2-58 chlorophenyl)-5-phenyl-1H-pyrazol-3-yl)phenoxy) butanoic acid]). It needs to be further determined to what extent the presence of MF1 alone, thought to be a ligand of fatty acid-binding protein 3 (FABP3), exerts any inhibitory effects on K+ currents, although the presence of MF1 alone did not affect GABA currents in this study. This issue is an important off-target effect and quite relevant to the present study; hence, it needs to be stated in the Discussion section of the revised manuscript.
  • In Figure 1D, the strength of GABA currents induced by MF at a concentration of 1 mM tended to be greater than that at a concentration of 10 m The results were somewhat different from that shown in Figure 1C. Please explain the reason appropriately.
  • Are the effects of MF1 on GABA currents linked to its binding FABP3? Please provide the evidence.
  • Is it possible that the benzodiazepine recognition site, to which the MF1 molecule bind, shares the structural or functional similarity to FABP3? Please comment on this issue properly.

Minor comments:

  • In lines 91 and 92, the text should be changed to “1 and 10 mM” …..
  • In lines 116-117, the sentence needs to be corrected.
  • In Table1, please add the full name of GTCs (i.e., generalized tonic convulsions).
  • In Table 1, MF1-induced shortening in GTCs onset time could be likely associated with the inhibition of potassium currents.
  • In line 247, the text should be changed to “2 mM” Mg-ATP.
  • In lines 252-253, the sentence needs to be rephrased.
  • Please use abbreviated name MF-1 appropriately. MF-1 (4-(2-(1-(2-58 chlorophenyl)-5-phenyl-1H-pyrazol-3-yl)phenoxy) butanoic acid) seems to be too short. MF-1 should add to the Abbreviation section of the manuscript.

Author Response

Major points:

1) Previous reports have demonstrated that benzodiazepine (e.g., midazolam or diazepam) might be capable of modulating the amplitude and gating of delayed rectifier K+ currents and that the underlying inhibitory action was not reversed by further addition of flumazenil (Vonderlin et al., Drug Des Devel Ther 2014;8:2263-2271; So et al., Eur J Pharmacol 2014;724:152-160). The delayed rectifier K+ currents are present in Neuro2A cells and they are highly likely to be inhibited by benzodiazepine derivatives (e.g., MF1, [4-(2-(1-(2-58 chlorophenyl)-5-phenyl-1H-pyrazol-3-yl)phenoxy) butanoic acid]). It needs to be further determined to what extent the presence of MF1 alone, thought to be a ligand of fatty acid-binding protein 3 (FABP3), exerts any inhibitory effects on K+ currents, although the presence of MF1 alone did not affect GABA currents in this study. This issue is an important off-target effect and quite relevant to the present study; hence, it needs to be stated in the Discussion section of the revised manuscript.

Ans. According to reviewer’s comment, we described the possibility that MF1 may affect electrophysiological properties as followed “Benzodiazepine drugs act on not only GABAA receptor but also delayed rectifier K+ channels. Midazolam behaves open-channel inhibitor with rapid onset of block and without frequency dependence in blocking of delayed rectifier K+ channels, thereby inhibiting amplitude of action potentials in NSC-34 neuronal cells [30, 31]. This effect is not blocked by flumazenil [31], suggesting that benzodiazepine receptor agonist has a possibility to suppress neuronal excitabilities in vivo independently with GABAergic action. As MF1 here shows benzodiazepine agonist-like effect, it may also affect electrophysiological properties of delayed rectifier K+ channels. We will try to investigate the effect of MF1 on delayed rectifier K+ currents in the future study.”. (Line 182 to 189)

2) In Figure 1D, the strength of GABA currents induced by MF at a concentration of 1 μM tended to be greater than that at a concentration of 10 μM. The results were somewhat different from that shown in Figure 1C. Please explain the reason appropriately.

Ans. The effect of GABA (1-100 μM) or MF1 (1-100 μM) alone on GABAA currents is shown in Figure 1A, B and C. On the other hand, the effect of GABA (1 or 10 μM) with or without MF1 (1 μM) and/or flumazenil (10 μM) on GABAAcurrents is exhibited in Figure 1D and E. We previously demonstrated that EC50 is 7.78 ± 0.95μM for GABAA receptor at the same plasmid as present study (J Pharmacol Sci. 2013, 121, 84–87.). Therefore, MF1 may show high sensitivity for low dose (1 μM) more than high dose of GABA-evoked currents.

3) Are the effects of MF1 on GABA currents linked to its binding FABP3? Please provide the evidence.

Ans. Since MF1-promoted GABAA currents is likely so rapid response (Figure 1D, E) and FABP3 is little or none expressed in endogenous neuro2A cells (Brain research. 2019, 1707, 190–197.), the effect of MF1 on GABA currents may be independent on its binding FABP3. On the other hand, it is unclear whether FABP3 interacts with GABAAreceptor and affects its function. In the future, we will try to reveal above using FABP3 and GABAA receptor co-expressed cells with or without MF1 treatment.

4) Is it possible that the benzodiazepine recognition site, to which the MF1 molecule bind, shares the structural or functional similarity to FABP3? Please comment on this issue properly.

Ans. We did not find any report about functional similarity between benzodiazepine recognition site in GABAA receptor and FABP3. Using BLAST, we compared the sequences of FABP3 to benzodiazepine recognition site binding proteins (benzodiazepine receptor-associated protein 1 and acyl-CoA-binding protein), however, there are no similar sequences between each protein (data not shown). Therefore, the benzodiazepine recognition site in GABAA receptor may not be similar to FABP3 in the structure and function. On the other hand, docosahexaenoic acid and fatty acid amides with an affinity for FABPs likely influences GABAA receptor function (Biochemistry. 2006, 45, 13118–13129; Pharmaceuticals (Basel). 2020, 13, 43.). Further studies are necessary to reveal the relationship between GABAA receptor, FABPs and fatty acids.

Minor points.

5) In lines 91 and 92, the text should be changed to “1 and 10 μM”.

Ans. According to reviewer’s comment, we corrected above space error. (Line 98 and 99)

6) In lines 116-117, the sentence needs to be corrected.

Ans. According to reviewer’s comment, we corrected the concerned sentence as followed “The anti-epileptic effect of MF1 was also investigated using PTZ-treated mice. DZP (5.0 mg/kg, i.p.) significantly suppressed the onset time of generalized tonic-clonic seizures (GTCs) and mortality as compared to vehicle-administered PTZ-treated mice.”. (Line 123 to 125)

7) In Table1, please add the full name of GTCs (i.e., generalized tonic convulsions).

Ans. According to reviewer’s comment, we added the full name of GTCs as generalized tonic-clonic seizures in legend of Table 1. (Line 129)

8) In Table 1, MF1-induced shortening in GTCs onset time could be likely associated with the inhibition of potassium currents.

Ans. Thank you for your comment. In the future study, we try to reveal the effect of MF1 on delayed rectifier K+channels.

9) In line 247, the text should be changed to “2 mM” Mg-ATP.

Ans. According to reviewer’s comment, we corrected above space error. (Line 261)

10) In lines 252-253, the sentence needs to be rephrased.

Ans. According to reviewer’s comment, we corrected the concerned sentence as followed “Membrane potential was clamped at -60 mV. Measured GABA currents were normalized to membrane capacitance in each cell.”. (Line 266 to 267)

11) Please use abbreviated name MF-1 appropriately. MF-1 (4-(2-(1-(2-58 chlorophenyl)-5-phenyl-1H-pyrazol-3-yl)phenoxy) butanoic acid) seems to be too short. MF-1 should add to the Abbreviation section of the manuscript.

Ans. According to reviewer’s comment, we added MF1 to list of abbreviation. (Line 310)

Reviewer 3 Report

 The authors reported that the  fatty acid-binding protein 3 (FABP3) ligand MF1 promotes GABA currents in  GABAA receptor-expressing cells and attenuates  pilocarpine-induced seizures in mice. The anticonvulsant effects of MF1 were blocked by treatment with flumazenil, suggesting that MF1 enhances  GABAA receptor function directly via the benzodiazepine recognition site. On the other hand, MF1 failed to affect pentylenetetrazol-induced seizures. The obtained data are original and fairly interesting. The methods are sound, however there are some concerns about doses of MF1. Generally, this paper has been well written.

Specific remarks:

Introduction

  1. “Most anticonvulsants promote GABAergic neuronal transmission by enhancing activity of the GABA type-42 A (GABAA) receptor and increasing GABA content in the synaptic cleft [4, 10]”. This is rather a controversial statement because voltage dependent sodium channel blockers are equally important class of antiepileptic drugs.
  2. “Our off-target analysis showed that MF1 has an affinity for the GABAA receptor, but its the effect on the GABAergic system remains unclear”. What exactly is the affinity (Kd) for the GABA A receptors? To which subunits of GABA A the compound does bind? Is there a preference  for binding of MF1 to a specific subunit configuration of GABA A receptor? Some answers to these questions  can be found in the first part of the Discussion. However, I feel that it would be better to transfer them to Introduction.

Results

  1. The sentence “Since benzodiazepine regents enhance GABA currents..,” should be corrected.
  2. “The anti-epileptic effect of MF1 was also analyzed in PTZ-treated mice. DZP (5.0 mg/kg, i.p.) significantly attenuated the onset time of generalized tonic-clonic seizures (GTCs) and mortality. However, MF1 had no effect in PTZ-treated mice (Table 1), suggesting that the anti-epileptic effect of MF1 (0.1 - 1.0 mg/kg, p.o.) may be milder than that of DZP (5.0 mg/kg, i.p.)”. It is unclear to me why the compunds were compared using different doses and routes of administration. Why the highest dose of MF1 was 1 mg/kg? Does it produce in higher doses some undesired effects? If not, this study should be supplemented with additional data on effect of MF1 in dose of 5 mg/kg on PTZ and PILO seizures.
  3. Does MF1 prevent pilocarpine seizures–related brain damage? Was histological analysis performed?
  4. Does flumazenil alone affect pilocarpine seizures?

Discussion

1.”Hence, chronic administration of MF1 also may promote GABAergic system activity in the cortex by increasing GAD67 expression. We plan to assess the chronic effects of MF1 on GAD67 levels in the future”. Does MF1 enhance GABA content in the brain?

  1. “Since chronic administration (at least once a day for seven consecutive weeks) of MF1 (1.0 mg/kg, p.o.) did not affect motor and cognitive function in mice [19, 22], MF1 may be a safer alternative for long-term therapy”. This observation does not exclude that long-term MF1 administration can lead to development of tolerance.

Author Response

1) “Most anticonvulsants promote GABAergic neuronal transmission by enhancing activity of the GABA type-42 A (GABAA) receptor and increasing GABA content in the synaptic cleft [4, 10]”. This is rather a controversial statement because voltage dependent sodium channel blockers are equally important class of antiepileptic drugs.

Ans. Thank you for your comments. According to reviewer’s comment, we toned down this sentence as followed “Many anticonvulsants promote GABAergic neuronal transmission by enhancing activity of the GABA type-A (GABAA) receptor and increasing GABA content in the synaptic cleft [4, 10].”. (Line 41 to 43)

2) “Our off-target analysis showed that MF1 has an affinity for the GABAA receptor, but its the effect on the GABAergic system remains unclear”. What exactly is the affinity (Kd) for the GABAA receptors? To which subunits of GABAA the compound does bind? Is there a preference for binding of MF1 to a specific subunit configuration of GABAA receptor? Some answers to these questions can be found in the first part of the Discussion. However, I feel that it would be better to transfer them to Introduction.

Ans. According to reviewer’s comment, we transferred GABAA receptor information form the discussion to introduction session as followed “The GABAA receptor complex consists of five subunits of different families (α1–6, β1–4, γ1–4, δ, ϵ and π) [12]. In the mammalian brain, the αβγ-subunit-containing complex is the major component of GABAA receptor; a minor complex with αβδ subunits is also observed in the hippocampal dentate gyrus, thalamus and cerebellum [13].GABAA receptor function is dependent on the type of subunit complex. For instance, the αβγ complex is sensitive to benzodiazepine, but this is not so for the αβδ complex [12, 13]. The α1βγ-type GABAA receptors are localized in the post synaptic region, while α4βδ-type GABAA receptors are observed in extra-synaptic region [14, 15].” (Line 47 to 53). We would like to investigate Kd value of MF1 for GABAA receptor in the future study.

3) The sentence “Since benzodiazepine regents enhance GABA currents,” should be corrected.

Ans. According to reviewer’s comment, we corrected the concerned sentence as followed “Since benzodiazepines facilitate GABAA currents without evoking them [27], we speculated that MF1 promotes GABAA currents in a manner similar to benzodiazepines.”. (Line 87 to 89)

4) “The anti-epileptic effect of MF1 was also analyzed in PTZ-treated mice. DZP (5.0 mg/kg, i.p.) significantly attenuated the onset time of generalized tonic-clonic seizures (GTCs) and mortality. However, MF1 had no effect in PTZ-treated mice (Table 1), suggesting that the anti-epileptic effect of MF1 (0.1 - 1.0 mg/kg, p.o.) may be milder than that of DZP (5.0 mg/kg, i.p.)”. It is unclear to me why the compunds were compared using different doses and routes of administration. Why the highest dose of MF1 was 1 mg/kg? Does it produce in higher doses some undesired effects? If not, this study should be supplemented with additional data on effect of MF1 in dose of 5 mg/kg on PTZ and PILO seizures.

Ans. We decided the dose and treatment routes of DZP (5.0 mg/kg, i.p.) as positive control (Neurosci Lett. 2012, 523, 115–118.). That of MF1 (0.1-1.0 mg/kg, p.o.) was also defined by previous our observations (Neuropharmacology. 2019,150, 164–174.; Int J Mol Sci. 2020, 21, 2230.). MF1 concentration in the mouse brain reaches 250 nM by oral administration of MF1 (1.0 mg/kg, p.o.) and Kd value for FABP3 is 302.8 ±â€¯130.3 nM (Brain research. 2019, 1707, 190–197.). Therefore, we suggest that MF1 (1.0 mg/kg, p.o.) may be so enough to function in the brain.

5) Does MF1 prevent pilocarpine seizures–related brain damage? Was histological analysis performed?

Ans. In the future study, we would like to try to investigate the effect of MF1 on PILO-induced hippocampal pyramidal and GABAergic neuronal damages.

6) Does flumazenil alone affect pilocarpine seizures?  

Ans. Thank you for your comments. According to Costa JP et al, flumazenil (25 mg/kg, i.p.) alone does not affect pilocarpine-induced seizures (Neurosci Lett. 2012, 523, 115–118.). We described it as followed “Doses of DZP (5.0 mg/kg, i.p.) and flumazenil (25 mg/kg, i.p.) were chosen, since these doses are enough to work in vivo without influence on normal animal condition [28].” (Line 244 to 246)

7) ”Hence, chronic administration of MF1 also may promote GABAergic system activity in the cortex by increasing GAD67 expression. We plan to assess the chronic effects of MF1 on GAD67 levels in the future”. Does MF1 enhance GABA content in the brain?

Ans. Thank you for your comments. Yes, we speculates upregulation of GABA contents in the mouse brain when MF1 increase GAD67 levels. We will investigate too in the future study.

8) “Since chronic administration (at least once a day for seven consecutive weeks) of MF1 (1.0 mg/kg, p.o.) did not affect motor and cognitive function in mice [19, 22], MF1 may be a safer alternative for long-term therapy”. This observation does not exclude that long-term MF1 administration can lead to development of tolerance.

Ans. Thank you for your comments. According to reviewer’s comment, we corrected our speculation as followed “Chronic administration (at least once a day for seven consecutive weeks) of MF1 (1.0 mg/kg, p.o.) did not affect motor and cognitive function in mice [23, 26]. Whereas tolerance and withdrawal of MF1 should be analyzed, we hope MF1 may be a safer alternative for long-term therapy.” (Line 215 to 218)

Round 2

Reviewer 2 Report

Minor comments:

Lines 129-130, Table 1 needs to be corrected. Dose in first column should be “5 mg/kg, 0.1 mg/kg, 0.3 mg/kg). “p.o.” should have the full name for clarity.

Author Response

Reviewer 2

Lines 129-130, Table 1 needs to be corrected. Dose in first column should be “5 mg/kg, 0.1 mg/kg, 0.3 mg/kg). “p.o.” should have the full name for clarity.

Answer. According to reviewer’s comment, we corrected above space errors and showed the full name of “i.p.” and “p.o.” as followed “i.p.: intraperitoneal, p.o.: per os.”. (Line 129 to 130).

Thank you